# Transcriptional Insights into Key Genes and Pathways Underlying Muscovy Duck Subcutaneous Fat Deposition at Different Developmental Stages

**DOI:** 10.3390/ani11072099

**Published:** 2021-07-15

**Authors:** Liping Guo, Congcong Wei, Li Yi, Wanli Yang, Zhaoyu Geng, Xingyong Chen

**Affiliations:** College of Animal Science and Technology, Anhui Agricultural University, No. 130 Changjiang West Road, Hefei 230036, China; guoliping@ahau.edu.cn (L.G.); 17681322537@163.com (C.W.); m15805656402@163.com (L.Y.); 18435131403@163.com (W.Y.); gzy@ahau.edu.cn (Z.G.)

**Keywords:** subcutaneous fat, RNA-seq, different developmental stages, Muscovy duck

## Abstract

**Simple Summary:**

Subcutaneous fat is an important factor affecting the meat quality and feed conversion rate of waterfowl. The current study compared the transcriptome data of Muscovy duck subcutaneous fat among three developmental stages, aiming at exploring the key regulatory genes for subcutaneous fat deposition. The results generated abundant candidate genes and pathways involving in subcutaneous fat deposition in Muscovy duck. This study provides an important reference for revealing the developmental mechanisms of subcutaneous fat in duck.

**Abstract:**

Subcutaneous fat is a crucial trait for waterfowl, largely associated with meat quality and feed conversion rate. In this study, RNA-seq was used to identify differentially expressed genes of subcutaneous adipose tissue among three developmental stages (12, 35, and 66 weeks) in Muscovy duck. A total of 138 and 129 differentially expressed genes (DEGs) were identified between 35 and 12 weeks (wk), and 66 and 35 wk, respectively. Compared with 12 wk, subcutaneous fat tissue at 35 wk upregulated several genes related to cholesterol biosynthesis and fatty acid biosynthesis, including *HSD17B7* and *MSMO1*, while it downregulated fatty acid beta-oxidation related genes, including *ACOX1* and *ACSL1*. Notably, most of the DEGs (92.2%) were downregulated in 66 wk compared with 35 wk, consistent with the slower metabolism of aging duck. Protein network interaction and function analyses revealed *GC*, *AHSG*, *FGG*, and *FGA* were the key genes for duck subcutaneous fat from adult to old age. Additionally, the PPAR signaling pathway, commonly enriched between the two comparisons, might be the key pathway contributing to subcutaneous fat metabolism among differential developmental stages in Muscovy duck. These results provide several candidate genes and pathways potentially involved in duck subcutaneous fat deposition, expanding our understanding of the molecular mechanisms underlying subcutaneous fat deposition during development.

## 1. Introduction

Poultry is one of the most important meat resources all over the world, providing consumers with diversified animal products such as high-quality meat, eggs, and down feather. Muscovy duck (*Cairina moschata*) is an excellent meat-type fowl species, famous for its high growth rate, delicious meat, and relatively high fat content [1,2,3]. Fat deposition is a crucial economic trait for livestock and poultry, largely associated with meat quality and feed conversion rate (FCR) [4,5,6]. Moderate fat deposition could improve meat quality [6]. Nevertheless, excessive fat deposition not only reduces feed utilization efficiency and carcass yield, but also affects the reproductive performance of poultry [7].

In avian species, most fats are deposited in abdominal fat, subcutaneous fat, visceral fat, and intramuscular fat. There is a certain spatiotemporal sequence in the accumulation of poultry fat. It is generally accepted that the deposition of subcutaneous fat in poultry is the earliest, but the deposition of abdominal fat is the fastest [8]. Broilers have the highest amount of fat deposited in the abdomen. Nevertheless, compared with chickens, meat-type ducks have a stronger fat deposition capacity, which could deposit a large amount of fat in subcutaneous tissue [9]. Aging is associated with fat distribution and changes [10]. Generally, the fat deposition in abdominal adipose tissue increases, while the fat deposition in subcutaneous adipose tissue decreases with aging [11,12,13]. However, little is known about the molecular mechanisms of Muscovy duck subcutaneous fat deposition during growth and development.

In this study, RNA-seq was used to investigate the gene expression profiles of subcutaneous adipose tissue at three developmental stages (12 wk (juvenile), 35 wk (adult), and 66 wk (agedness)) in Muscovy duck. The result might lead to more knowledge on the molecular mechanism of subcutaneous fat deposition in duck.

## 2. Materials and Methods

### 2.1. Ethics Statement

All experimental procedures and samples’ collection were performed according to the regulations for the Administration of Affairs Concerning Experimental Animals of the State Council of the People’s Republic of China and the Institutional Animal Care and were approved by the Institutional Animal Care and Use Committee of the College of Animal Science and Technology, Anhui Agricultural University, Hefei, China (permit No. AHAU20101025).

### 2.2. Animals and Adipose Tissue Sampling

Healthy Muscovy ducks were obtained from Anqing Yongqiang Agricultural Science and Technology Co. Ltd. (Anqing, China), China. All ducks (*n* = 27) were raised according to standard procedure. Three developmental stages, including 12 wk, 35 wk, and 66 wk, were explored in this study. At each stage, 9 ducks (female) with similar weights were selected. After slaughter, subcutaneous adipose tissue was collected and stored in liquid nitrogen.

### 2.3. RNA Extraction and RNA Sequencing

Total RNA was extracted from 27 samples using the TRIzol reagent (Invitrogen Corp., Carlsbad, CA, USA) according to the manufacturer’s instructions. The RNA concentration and integrity were assessed using a 2100 Bioanalyzer and RNA 6000 Nano Kit (Agilent Technologies, Santa Clara, CA, USA) separately. RNA samples with an OD260 nm/OD280 nm ratio between 1.9 and 2.0 were used for RNA sequencing. Before sequencing, 3 samples were quantitatively mixed to form 3 biological replicates at each stage. Based on ultra-high-throughput sequencing (HiSeq 2500; Illumina, San Diego, CA, USA), RNA-sequencing was performed by Shanghai Biotechnology Corporation (Shanghai, China).

### 2.4. Bioinformatics Analysis of RNA-Seq

Sequence adapters and low-quality reads were removed using Trimmomatic [14]. The quality control of the raw sequence was performed by FastX (http://www.bioinformatics.bbsrc.ac.uk/projects/fastqc/ (accessed on 2 August 2020)). The paired-end reads were aligned to the duck reference genome using Tophat2 version 2.0.11 (http://ccb.jhu.edu/software/tophat/index.shtml (accessed on 2 August 2020)). To calculate the expression quantity of each transcript, alignment results were analyzed by Cufflinks version 2.1.1. The FPKM (Fragments Per Kilobase of exon model per Million mapped reads) method was used to quantify the gene expression level of all samples. Differentially expressed genes (DEGs) were identified using the DESeq [15] (2012) R package functions estimate SizeFactors and nbinomTest. *p* value < 0.05 and fold change (FC) > 1.5 or FC < 0.67 was set as the threshold for significantly differential expression.

Hierarchical clustering analyses were performed based on all DEGs. A Venn diagram was made by the online software jvenn (http://jvenn.toulouse.inra.fr/app/example.html (accessed on 2 August 2020)). Gene ontology (GO) analysis of DEGs was performed using DAVID functional annotation. The Kyoto Encyclopedia of Genes and Genomes (KEGG) pathway enrichment analysis was performed by KOBAS 3.0 (http://kobas.cbi.pku.edu.cn (accessed on 2 August 2020)). The significance level for GO terms and the KEGG pathway was set with the *p* value < 0.05. The protein–protein interaction (PPI) network was constructed by STRING online database (https://string-db.org/ (accessed on 2 August 2020)) and visualized using Cytoscape _v3.8.0 software [16]. The Molecular Complex Detection (MCODE) [17] application in Cytoscape was used to screen the modules of the PPI network.

## 3. Results

### 3.1. Overall Assessment of RNA-Seq Data

RNA-seq data were obtained from subcutaneous adipose samples of 12 wk, 35 wk, and 66 wk in Muscovy duck. Sequencing of subcutaneous fat yielded 46.2 million raw reads per sample on average. After filtering, the number of clean reads ranged from 38,826,006 to 57,268,052, with the percentage of clean reads above 92.99%. The Q20 Value ranged from 88.66% to 93.64%. An average of 41.23% reads was mapped to the reference genome (Table 1).

### 3.2. DEGs Analysis among Three Developmental Stages

In total, 14,519 genes were detected in the 9 sequencing samples of Muscovy duck subcutaneous adipose tissue. A total of 138 DEGs were identified between subcutaneous adipose tissue of 35 wk and 12 wk, including 100 known DEGs and 38 novel genes. Coincidentally, half of the DEGs were upregulated in 35 wk (Figure 1A and Appendix A). Compared with 35 wk, 129 DEGs were identified in 66 wk, of which 119 DEGs were downregulated (Figure 1B and Appendix A). Moreover, 17 DEGs were both significantly differentially expressed in both comparisons (Figure 1C). In addition, from the hierarchical clustering result, we can see samples with the same developmental stage clustered together. These results suggested that there were obvious differences between different developmental stages of subcutaneous adipose tissue in Muscovy duck.

The list of the top 20 known up-and down-regulated DEGs between the two comparisons, ranked by FC, are shown in Table 2 and Table 3. The most altered gene between 35 wk and 12 wk was *PCK1* (downregulated). The most altered gene between 66 wk and 35 wk was *MB* (downregulated).

### 3.3. Functional Enrichment of the DEGs between 35 wk and 12 wk

The function of the known DEGs was examined by GO enrichment analysis. A total of 20 GO terms were assigned between 35 wk and 12 wk subcutaneous adipose tissue, and each term contained more than two DEGs (Figure 2A). Among them, six biological processes (BP) terms, one cellular component (CC), and five molecular functions (MF) were significantly enriched (Appendix A) (*p* < 0.05).

After KEGG pathway analysis, six pathways were significantly enriched between 35 wk and 12 wk subcutaneous adipose tissue (Figure 2B) (*p* < 0.05). Most of the enriched pathways were closely related to lipid metabolism, including steroid biosynthesis, the adipocytokine signaling pathway, the PPAR signaling pathway, metabolic pathways, and fatty acid degradation. Further analysis revealed that seven DEGs (*ACOX1*, *ACSL1*, *CYP2R1*, *HSD17B7*, *MSMO1*, *PCK1*, and *PRKCQ*) were enriched in the PPAR signaling pathway, steroid biosynthesis, fatty acid degradation, and the adipocytokine signaling pathway.

### 3.4. PPI Network Analysis of DEGs between 35 wk and 12 wk

To further investigate the interaction of DEGs and screen the hub genes related to lipid metabolism, the PPI network analysis was performed. Based on the STRING online database and Cytoscape software, the network of DEGs between 35 wk and 12 wk was built, consisting of 48 nodes and 63 edges (Figure 3A). To further screen the hub genes associated with lipid deposition, module analysis was subsequently performed using the cytoHubba. The top 10 nodes ranked by the MCC method and the related DEGs formed the other core sub-network (Figure 3B). The network was mainly divided into two parts. Consistent with the KEGG enrichment results, DEGs related to lipid metabolism, including *ACOX1*, *ACSL1*, *PCK1*, *MSMO1*, and *HSD17B7* were closely linked to form a network module. Moreover, Sterol regulatory element binding transcription factor 2 (*SREBP2*), closely linked to *ACOX1*, *ACSL1*, *PCK1,* and *MSMO1*, were also the hub gene. In addition, several heat shock protein family members, including HSPA2, HSP90AA1, HSPA5, HSPA4, and HSPA4L, constituted the other network module.

The three significant modules, including module 1 (MCODE score = 5.6), module 2 (MCODE score = 3), and module 3 (MCODE score = 3), were constructed from the PPI network of DEGs between 35 and 12 weeks using MCODE (Figure 4). Module 1 (Figure 4A) included 6 nodes and 14 edges. Module 2 (Figure 4B) included three nodes and three edges. Module 3 (Figure 4C) included three nodes and three edges.

### 3.5. Functional Enrichment Analysis of the DEGs between 66 wk and 35 wk

A total of 26 BP, 11 CC, and 4 MF terms were significantly enriched between subcutaneous adipose tissue of 66 wk and 35 wk (Appendix A) (*p* < 0.05). The top 20 GO terms are shown in Figure 5A. As expected, some enriched terms were related to lipid metabolism, such as the BP term “cholesterol homeostasis” and the CC term “very-low-density lipoprotein particle”. CYP7A1, ANGPTL3, and APOB were significantly enriched in cholesterol homeostasis (GO:0042632). Additionally, APOH and APOB were significantly enriched in very-low-density lipoprotein particle terms (GO:0034361). All the abovementioned genes were significantly downregulated in 66 wk subcutaneous fat, indicating that cholesterol metabolism of subcutaneous fat in 66 wk was reduced more than that of 35 wk.

Based on 78 known DEGs, nine KEGG pathways were significantly enriched (Figure 5B) (*p* < 0.05). As expected, some pathways related to lipid metabolism were enriched, such as the PPAR signaling pathway. Three DEGs (*CYP7A1*, *ACSL5,* and a new gene) were enriched in the PPAR signaling pathway, which were significantly downregulated in 66 wk subcutaneous adipose tissue, indicating that lipid metabolism of subcutaneous adipose tissue in 66 wk was reduced more than that of 35 wk. Moreover, several metabolism-related pathways were enriched, including Tyrosine metabolism, Fructose and mannose metabolism, phenylalanine metabolism, and pentose phosphate pathway. In addition, the calcium signaling pathway and the tight junction pathway were also enriched.

### 3.6. PPI Network Analysis of DEGs between 66 wk and 35 wk

The PPI network of DEGs between 66 wk and 35 wk was performed. A total of 63 nodes and 368 edges were identified (Figure 6A). To further screen the hub genes, module analysis was performed using the cytoHubba. Ranked by the MCC method, the top 10 nodes and the related DEGs formed the other core sub-network (Figure 6B). The top 10 genes were *GC*, *AHSG*, *FGG*, *FGA*, *SPP2*, *CPB2*, *ITIH3*, *FETUB*, *FTCD,* and *TM4SF4*. These hub genes were closely linked to lipid metabolism genes *CYP7A1*, *ANGPTL3*, *APOB*, *APOH*, and *ACSL5*, which may be the key genes regulating lipid metabolism of Muscovy duck subcutaneous adipose tissue from 35 wk to 66 wk.

The three significant modules, including module 1 (MCODE score = 16.5), module 2 (MCODE score = 12.7), and module 3 (MCODE score = 5.6), were constructed from the PPI network of DEGs between 66 and 35 weeks using MCODE (Figure 7). Module 1 (Figure 7A) included 17 nodes and 132 edges. Module 2 (Figure 7B) included 13 nodes and 76 edges. Module 3 (Figure 7C) included 6 nodes and 14 edges.

## 4. Discussion

Duck meat consumption has been on the rise worldwide. Fat deposition is one of the key factors affecting the meat quality of poultry [18,19]. In this study, we compared the transcriptome data of subcutaneous fat in Muscovy ducks at different development stages.

Around 46.2 million raw reads were obtained from each sample and the clean reads ratio was above 92.99%. The mapping rate of this study was 41.23% on average, relatively low compared with chicken, Pekin Duck, and other poultry species [20,21]. Zeng’s research (2015) indicated that the mapping rate of the sequences of Muscovy duck liver was 44%, which is well in agreement with our study [22]. The major reason is probably that Muscovy duck is distantly related to *Anas platyrhynchos* (the species of the reference genome). Muscovy duck is originated from *Cairina*
*moschata* [23], while other domestic duck originate from *Anas platyrhynchos* [24]. Thus, there is indeed a difference between their morphological and genomic characteristics. In addition, the sequence information of *Anas platyrhynchos* is not integral in the NCBI [25], which may be the other reason for the low mapping rate. Despite a lower mapping rate, numerous mapped reads could meet the subsequent analysis.

Transcriptome analysis reveals that there were significant differences in gene expression among different development stages. As the age increased from 12 wk to 35 wk, Muscovy duck reached maturity, accompanied by increased fat deposition. Several genes encoding key enzymes involved in lipid metabolism were expressed differentially. For example, hydroxysteroid 17-beta dehydrogenase 7 (HSD17B7) and methylsterol monooxygenase 1 (MSMO1) were involved in cholesterol biosynthesis and fatty acid biosynthesis [26,27], which were both significantly upregulated in 35 wk subcutaneous adipose tissue, indicating that cholesterol biosynthesis of subcutaneous adipose tissue in 35 wk was more enhanced than that of 12 wk. Acyl-coa oxidase 1 (ACOX1) and acyl-CoA synthetase long-chain family member 1 (ACSL1), significantly enriched in the PPAR signaling pathway and Fatty acid degradation, were related to fatty acid beta-oxidation [28], which were both significantly downregulated in 35 wk subcutaneous adipose tissue, indicating that fatty acid beta-oxidation of subcutaneous adipose tissue in 35 wk was more reduced than that of 12 wk. Phosphoenolpyruvate carboxykinase 1 (PCK1), commonly considered the control point for gluconeogenesis [29], was significantly downregulated in 35 wk subcutaneous adipose tissue, indicating that gluconeogenesis of subcutaneous adipose tissue in 35 wk was reduced. Moreover, PCK1 is also involved in lipid and carbohydrate metabolism [30].

It is noticeable that most of the DEGs (92.2%) were downregulated in 66 wk compared to 35 wk, indicating that the total metabolism of subcutaneous adipose tissue in 66 wk was slower than that of 35 wk. This could be due to the age of 35 wk to 66 wk representing the stages of Muscovy duck from adult to aging.

Apolipoprotein B (APOB) is related to cholesterol uptake and lipids transport [31,32]. Apolipoprotein H (APOH) is involved in the activation of lipoprotein lipase in lipid metabolism. Previous studies have associated APOH polymorphism and variant with various lipid traits [33,34,35]. Acyl-CoA synthetase long-chain family member 5 (ACSL5), part of the class of acyl-CoA synthetases, is involved in fatty acid activation, transport and degradation, and lipid synthesis [36]. Ablation of ACSL5 significantly increased energy expenditure and delayed triglyceride absorption in mice [37].

Through PPI network analysis, *GC*, *AHSG,* and *FGA* were the hub genes between 66 wk and 35 wk of subcutaneous adipose tissue. Much evidence suggested that low vitamin D status was associated with adiposity [38,39,40]. The *GC* gene encodes the vitamin D binding protein, which can affect low-density lipoprotein cholesterol levels [41]. Additionally, a recent study showed that the GC haplotype was associated with plasma concentrations of high-density lipoprotein cholesterol, low-density lipoprotein cholesterol, and triglycerides in humans [42]. A previous study has shown that alpha 2-HS glycoprotein (AHSG) plasma levels were positively associated with liver fat accumulation in humans [43]. Moreover, Lavebratt’s study found that polymorphism of AHSG was associated with lipolysis of subcutaneous adipocytes [44]. Hence, the downregulated expression of *APOB*, *APOH*, *ACSL5*, *GC,* and *AHSG*, etc., may be associated with reduced lipid accumulation in the aging duck.

Based on the identified DEGs, KEGG pathway enrichment was performed to explore the regulatory network of lipid deposition underlying differential developmental stages in duck subcutaneous fat tissue. Comparing 35 wk with 12 wk of duck subcutaneous adipose tissue, six pathways were significantly enriched. In particular, five out of six pathways were directly related to lipid metabolism, including the PPAR signaling pathway, steroid biosynthesis, the adipocytokine signaling pathway, the metabolic pathways, and fatty acid degradation. Moreover, PPAR signaling, adipocytokine signaling, and fatty acid degradation pathways were linked through *ACSL1*, *PCK1*, and *ACOX1* genes. Whereas in the comparison of 66 wk and 35 wk duck subcutaneous fat, the PPAR signaling pathway was the only enriched pathway directly related to fat metabolism. It is well known that PPAR signaling is the most important pathway to regulate lipid metabolism [45,46,47]. Furthermore, the PPAR signaling pathway was the common pathway between the two comparisons, indicating that PPAR signaling might be the key pathway for lipid deposition in duck subcutaneous adipose tissue.

## 5. Conclusions

In summary, this study compared the transcriptome data of subcutaneous fat in Muscovy duck at different developmental stages. The results showed that subcutaneous fat tissue of 35 wk upregulated several genes related to cholesterol biosynthesis and fatty acid biosynthesis, while it downregulated fatty acid beta-oxidation-related genes. Notably, most of the DEGs were downregulated in 66 wk compared to 35 wk, consistent with the slower metabolism of aging duck. In addition, we identified several regulatory hub genes that potentially can be used in duck breeding. These findings provide new clues on exposing the molecular mechanisms underlying the regulating of Muscovy duck adipose tissue at different stages.

## Figures and Tables

**Figure 1 animals-11-02099-f001:**
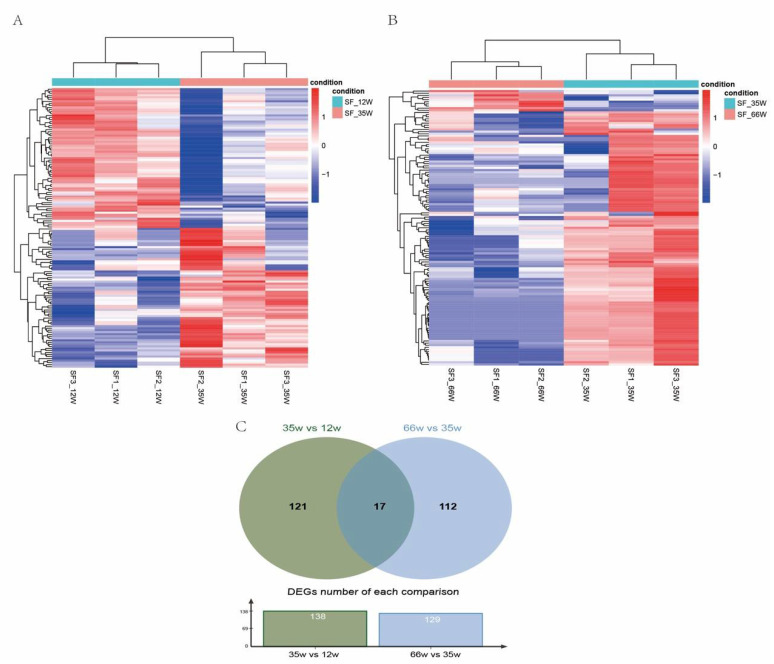
Differentially expressed genes (DEGs) of Muscovy duck subcutaneous fat (SF) at different developmental stages. (**A**). Hierarchical clustering analyses were performed based on DEGs between 35 and 12 weeks. (**B**). Hierarchical clustering analyses were performed based on DEGs between 66 and 35 weeks. (**C**). Venn diagram of DEGs at different developmental stages.

**Figure 2 animals-11-02099-f002:**
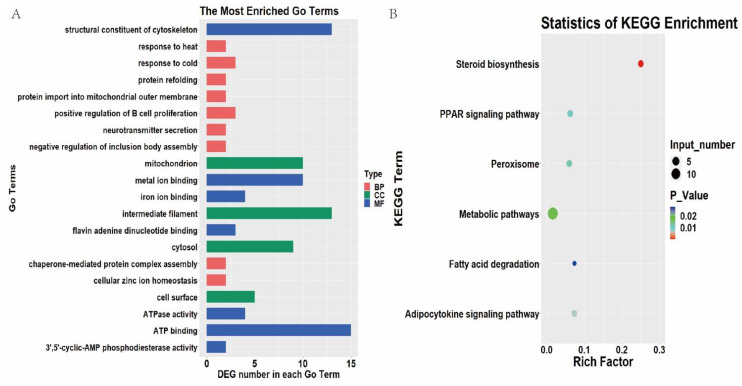
GO and KEGG enrichment of differentially expressed genes (DEGs) between 35 and 12 weeks. (**A**). Top 20 GO terms of DEGs between 35 and 12 weeks. (**B**). Enriched KEGG pathway of DEGs between 35 and 12 weeks.

**Figure 3 animals-11-02099-f003:**
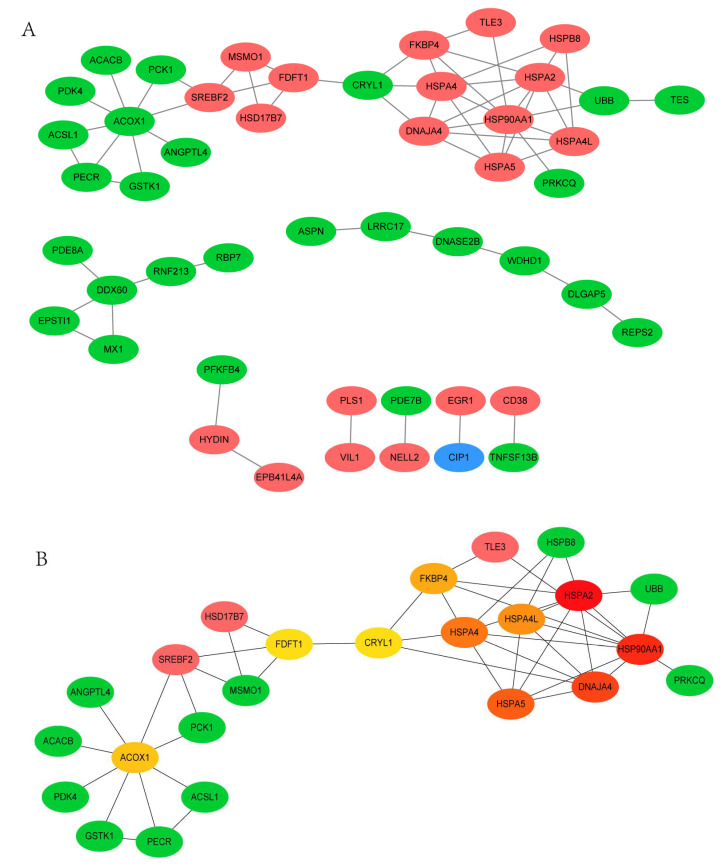
(**A**). PPI network analysis of differentially expressed genes (DEGs) between 35 and 12 weeks. A total of 48 nodes and 63 edges were screened. The green node represents the downregulated DEGs, and the red one represents the upregulated DEGs. (**B**). The color of nodes represented the MCC score of each gene. The color ranging from red to green represents high to low rank of DEGs.

**Figure 4 animals-11-02099-f004:**
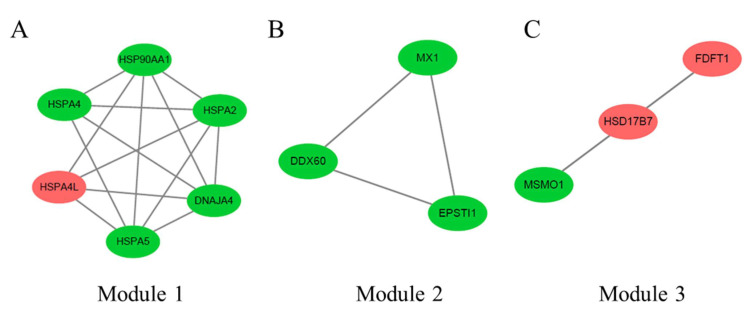
The three protein–protein interaction (PPI) hub network modules were constructed from the PPI network of DEGs between 35 and 12 weeks using MCODE. (**A**) Module 1 (MCODE score = 5.6), (**B**) module 2 (MCODE score = 3), and (**C**) module 3 (MCODE score = 3). The green node represents the downregulated DEGs, and the red one represents the upregulated DEGs.

**Figure 5 animals-11-02099-f005:**
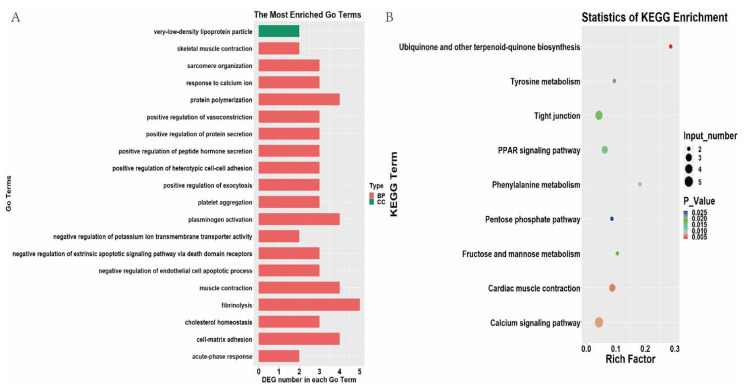
GO and KEGG enrichment of differentially expressed genes (DEGs) between 66 and 35 weeks. (**A**). Top 20 GO terms of DEGs between 66 and 35 weeks. (**B**). Enriched KEGG pathway of DEGs between 66 and 35 weeks.

**Figure 6 animals-11-02099-f006:**
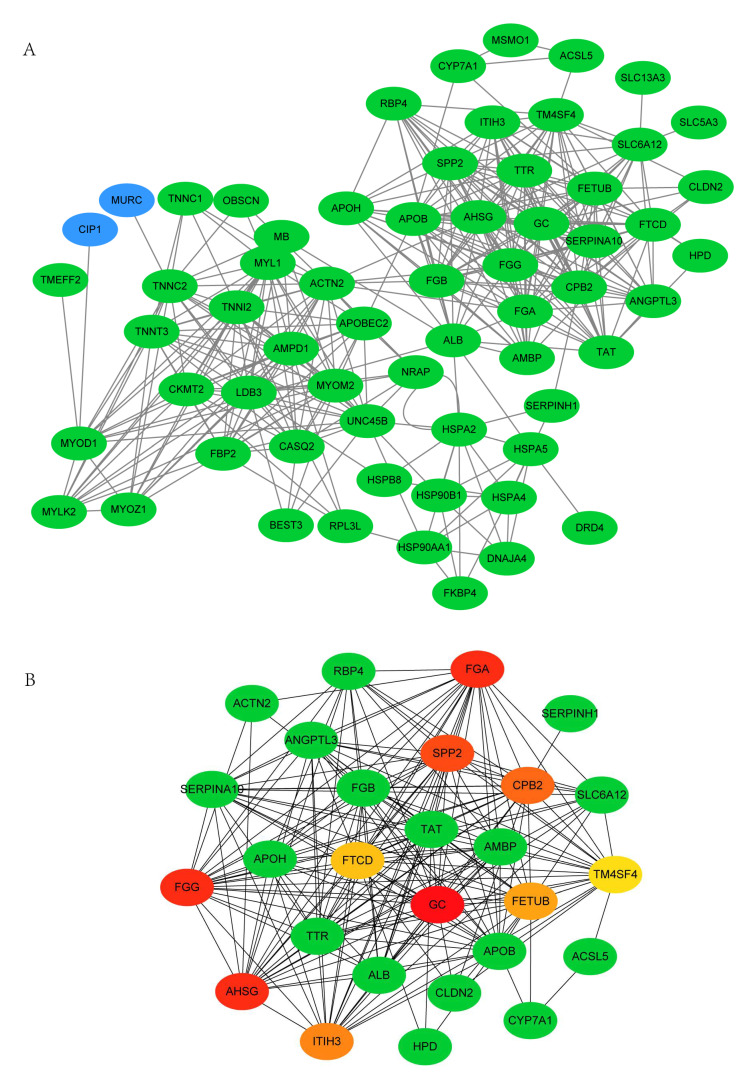
(**A**). PPI network analysis of differentially expressed genes (DEGs) between 66 and 35 weeks. A total of 63 nodes and 368 edges were screened. The green node represents the downregulated DEGs, and the blue one represents the related genes. (**B**). The color of nodes represented the MCC score of each gene. The color ranging from red to green represents high to low rank of DEGs.

**Figure 7 animals-11-02099-f007:**
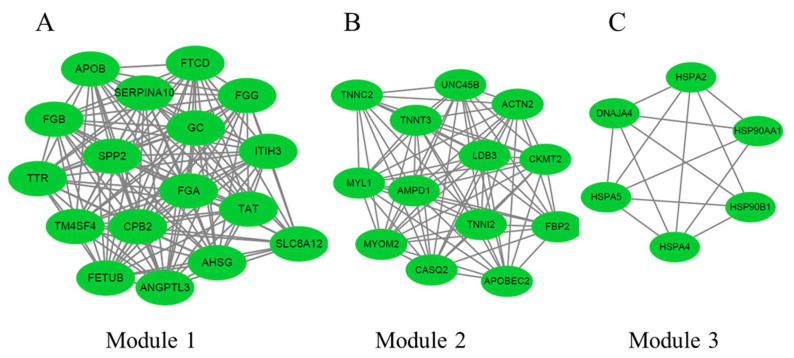
The three protein–protein interaction (PPI) hub network modules were constructed from the PPI network of DEGs between 66 and 35 weeks using MCODE. (**A**) Module 1 (MCODE score = 16.5), (**B**) module 2 (MCODE score = 12.7), and (**C**) module 3 (MCODE score = 5.6). The green node represents the downregulated DEGs.

**Table 1 animals-11-02099-t001:** Statistical information of RNA-seq data in each sample.

Sample	Raw Reads	Q20 Value	Clean Reads	Clean Reads Ratio	Mapped Reads	Mapping Ratio
35w-3	42,850,794	89.89%	40,327,910	94.11%	16,701,309	41.46%
35w-2	48,548,724	90.02%	45,637,136	94.00%	19,810,653	43.44%
35w-1	43,708,580	89.88%	41,001,746	93.81%	17,815,651	43.48%
66w-3	41,752,056	88.66%	38,826,006	92.99%	13,653,232	35.19%
66w-2	39,392,088	91.83%	37,119,504	94.23%	15,430,183	41.60%
66w-1	43,543,486	91.58%	40,983,242	94.12%	17,051,338	41.64%
12w-3	44,862,056	91.83%	42,440,164	94.60%	15,662,882	36.93%
12w-2	50,191,846	93.64%	47,533,720	94.70%	20,748,045	43.69%
12w-1	60,636,404	93.22%	57,268,052	94.45%	24,977,931	43.68%

**Table 2 animals-11-02099-t002:** The top 20 known differentially expressed genes between 35 wk and 12wk.

Gene Name	FC	*p* Value	Description	Trend
*PCK1*	0.03	0.03	Phosphoenolpyruvate carboxykinase	Down
*MX1*	0.16	0.04	Interferon-induced GTP-binding protein Mx isoform X1	Down
*TMPRSS7*	0.19	0.00	Transmembrane protease serine 7 isoform X1	Down
*EPSTI1*	0.20	0.01	Epithelial-stromal interaction protein 1	Down
*ANGPTL5*	0.21	0.01	Angiopoietin-related protein 5 isoform X4	Down
*NRXN1*	0.22	0.03	Neurexin-1 isoform X6	Down
*C18orf21*	0.24	0.01	Uncharacterized protein c18orf21-like protein, partial	Down
*ABHD6*	0.26	0.00	Monoacylglycerol lipase ABHD6	Down
*SYN3*	0.27	0.04	Synapsin-3	Down
*TRH*	0.27	0.01	Prothyroliberin, partial	Down
*NELL2*	16.93	0.00	Protein kinase C-binding protein NELL2, partial	Up
*MLKL*	15.35	0.00	Mixed lineage kinase domain-like protein, partial	Up
*HYDIN*	14.59	0.03	Belongs to the cytochrome P450 family	Up
*ASTN1*	10.16	0.04	Astrotactin-1	Up
*KIAA1211*	7.74	0.04	Uncharacterized protein KIAA1211 homolog isoform X2	Up
*GDF1*	5.25	0.02	Embryonic growth/differentiation factor 1	Up
*LCN15*	4.72	0.01	LOW QUALITY PROTEIN: lipocalin-15-like	Up
*PLS1*	4.60	0.04	Plastin-1	Up
*CD38*	4.55	0.04	ADP-ribosyl cyclase/cyclic ADP-ribose hydrolase 2	Up
*FGF12*	4.02	0.02	Fibroblast growth factor 12 isoform X3	Up

**Table 3 animals-11-02099-t003:** The top 20 known differentially expressed genes between 66 wk and 35wk.

Gene Name	FC	*p* Value	Description	Trend
*MB*	0.00	0.01	Myoglobin, partial	Down
*APOBEC2*	0.00	0.02	Apolipoprotein B mRNA editing enzyme catalytic subunit 2	Down
*HPD*	0.00	0.02	4-hydroxyphenylpyruvate dioxygenase	Down
*SPP2*	0.00	0.03	secreted phosphoprotein 24	Down
*FTCD*	0.00	0.03	Formimidoyltransferase-cyclodeaminase	Down
*CYP7A1*	0.00	0.03	Cytochrome P450 7A1, partial	Down
*TM4SF4*	0.00	0.04	transmembrane 4 L6 family member 4	Down
*SLC13A3*	0.00	0.04	Solute carrier family 13 member 3, partial	Down
*CPB2*	0.00	0.04	carboxypeptidase B2	Down
*SLC6A12*	0.00	0.05	sodium- and chloride-dependent betaine transporter	Down
*FGA*	0.00	0.01	fibrinogen alpha chain	Down
*FGB*	0.00	0.01	fibrinogen beta chain	Down
*APOB*	0.00	0.01	apolipoprotein B-100 isoform X1	Down
*ALB*	0.00	0.02	albumin	Down
*TCN2*	5.00	0.03	transcobalamin-2, partial	Up
*SYT12*	4.63	0.04	Synaptotagmin-12, partial	Up
*GNA14*	3.90	0.03	G protein subunit alpha 14	Up
*PAMR1*	3.44	0.03	inactive serine protease PAMR1 isoform X2	Up
*TNIP2*	3.26	0.02	TNFAIP3-interacting protein 2	Up
*NET1*	2.15	0.05	neuroepithelial cell-transforming gene 1 protein	Up

## Data Availability

The raw sequencing data reported in this paper have been deposited in the Genome Sequence Archive in BIG Data Center under accession number CRA004333 that can be publicly accessed at https://ngdc.cncb.ac.cn/gsa (accessed on 7 July 2021).

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
