# Peer review of "Transcriptional Insights into Key Genes and Pathways Underlying Muscovy Duck Subcutaneous Fat Deposition at Different Developmental Stages"

_animals, 2021, doi:10.3390/ani11072099_

Round 1

Reviewer 1 Report

The authors have been addressed most of my comments and the manuscript has been improved now. However, I could not verify the information regarding the analyses using the Deseq package. As mentioned in the response, the authors have been used the FDR, but in the manuscript, the writing is the p-value, not FDR. In fact, DEseq 2 provided both uncorrected p values and p values adjusted using BH (FDR). They also have to provide the raw results of  DE analyses in supplementary files which will be important for the readers. 

In addition, it is also important to upload the raw sequence data in the SRA deposit such as NCBI or ENSEMBL for reproducibility of analysis. 

Author Response

Dear reviewer,

I am very grateful to your valuable suggestions. Here are responses to the comments.

Question 1: The authors have been addressed most of my comments and the manuscript has been improved now. However, I could not verify the information regarding the analyses using the Deseq package. As mentioned in the response, the authors have been used the FDR, but in the manuscript, the writing is the p-value, not FDR. In fact, DEseq 2 provided both uncorrected p values and p values adjusted using BH (FDR). They also have to provide the raw results of DE analyses in supplementary files which will be important for the readers.

Response: After confirmation, the p-value was used in my study. P value and FDR are both parameters based on statistics, used to measure the significant level of gene expression. In comparison to p-value, FDR was more strict. But p-value was also commonly used in the literature [1-3]. In addition, the results of transcriptomic data mainly provide a reference for future research. Many fat deposition related genes and pathways were identified and enriched in my study, which could also prove the relatively reliability of the analysis.

Question 2: In addition, it is also important to upload the raw sequence data in the SRA deposit such as NCBI or ENSEMBL for reproducibility of analysis.

Response: the raw data for this study was submitted to the Genome Sequence Archive in BIG Data Center, Beijing Institute of Genomics (BIG), Chinese Academy of Sciences, under accession number CRA004333.

References

  1. Chen, B.; Liang, G.; Zhu, X.; Tan, Y.; Xu, J.; Wu, H.; Mao, H.; Zhang, Y.; Chen, J.; Rao, Y.; et al. Gene Expression Profiling in Ovaries and Association Analyses Reveal HEP21 as a Candidate Gene for Sexual Maturity in Chickens. Animals (Basel) 2020, 10, 181.
  2. Ye, P.; Ge, K.; Li, M.; Yang, L.; Jin, S.; Zhang, C.; Chen, X.; Geng, Z. Egg-laying and brooding stage-specific hormonal response and transcriptional regulation in pituitary of Muscovy duck (Cairina moschata). Poultry Science 2019, 98, 5287-5296.
  3. Yi, B.; Chen, L.; Sa, R.; Zhong, R.; Xing, H.; Zhang, H. High concentrations of atmospheric ammonia induce alterations of gene expression in the breast muscle of broilers (Gallus gallus) based on RNA-Seq. BMC Genomics 2016, 17, 598.

Reviewer 2 Report

The authors have addressed my concerns in an adequate manner. I have no further comments.

Author Response

Dear reviewer,

I am very grateful to your approval and valuable suggestions. Thank you very much!

Round 2

Reviewer 1 Report

Dear Authors, 

Thank you for your reponse. All my comments have been addressed. 

This manuscript is a resubmission of an earlier submission. The following is a list of the peer review reports and author responses from that submission.

Round 1

Reviewer 1 Report

The manuscript “Transcriptional insights into key genes and pathways underlying Muscovy duck subcutaneous fat deposition at different developmental stages” is important for understanding the genes and biological pathways underlying the development of duck. The study identified some key genes which have biological meaning in the context of lipid metabolisms. The study, however, has a limited sample size for each group. In addition, the methods need to be clarified as the authors did not provide any information regarding the bioinformatic analyses. Moreover, some justification regarding the p-value and DE analyses should be included.

Minors

Line 23: How many samples in each group.

Line 32: Defined the DEGs firstly, before using it

wk: it is not a common abbreviation, so the authors needed to define it two.

Line 26: Why did the authors choose the unadjusted p-values. Defined FC for fold change.

Line 45-46: please provide the supporting references.

Line 51-60: as the manuscript involving in a different growth state of the duck, the authors might summarize the growth stage of the duck and how the growth curve of the difference stage looks like.

Line 74-75: Why did the authors choose these weeks? Please justifying.

Are these animals related?

Are the any genetic/genomic selections for these animals or herds?

Line 77-79: As the fat amount is highly related to body weight, please provide the bodyweight of these animals. How did the authors select animals?

Please provide the details of bioinformatic analyses, the software for using the quality control, trimming, alignment, etc.

How did the authors perform DE analyses, which software? Did the authors normalize the data, which models used in DE analyses?

Why did the authors select p-value? The authors should use P values controlling for multiple testing such as FDR?

Line 100: Give citation for Cytoscape

The raw sequence data should be deposited in the public domain for the reproducibility of research.

Please avoid the abbreviation in the table and figure titles.

Table 1 is not so important. I suggest the authors present the table with the top DE genes for each comparison group.

Line 114: What is the reason for the selection of the log fold change threshold of 0.58 and please define the abbreviation for it

I suggested the authors used the Volcano plots instead of the heatmap in figure S1.

Line 135 and others: Please keep gene names in Italics.

Line 147: To identify the subnetwork, I suggest the authors use another app such as the MCODE app

Line 172: Change 3 to three and do the same for a number smaller than 11

Line 214-217: it is not necessary to repeat the results.

Line 236: please explore the relationship between HSP and fat deposition or lipid metabolism.

Line 239-240: The assumption is vague and no supporting references.

Line 242: the sentence is not supporting any things?  

Line 251-252 Explore more in the mechanism.

Line 258-259: in which species?

Line 261: How did the authors explain the link between the fat deposition in duck with the plasma lipid in humans?

Author Response

Dear editor,

I quite appreciate your favorite consideration and the reviewer’s insightful comments. Now I have revised the manuscript according to the reviewer’s comments. The revisions were addressed point by point below.

Reviewer 1:

The manuscript “Transcriptional insights into key genes and pathways underlying Muscovy duck subcutaneous fat deposition at different developmental stages” is important for understanding the genes and biological pathways underlying the development of duck. The study identified some key genes which have biological meaning in the context of lipid metabolisms. The study, however, has a limited sample size for each group. In addition, the methods need to be clarified as the authors did not provide any information regarding the bioinformatic analyses. Moreover, some justification regarding the p-value and DE analyses should be included.

Thank you for your comments and suggestions. Three samples per group were usually regarded as the minimum for statistical analysis. More samples will be designed in the future experiment. The information of bioinformatic analyses were added in the Materials and Methods section. Additionally, the p-value and DE analyses were discussed below.

Minors

Line 23: How many samples in each group.

Response: At each stage 9 ducks were selected. There were totally 27 ducks in three periods. Before sequencing, 3 RNA samples were quantitatively mixed to construct 3 pools at each stage.

Line 32: Defined the DEGs firstly, before using it

wk: it is not a common abbreviation, so the authors needed to define it too.

Response: The DEGs was defined in line 25. The wk was defined in line26.

Line 26: Why did the authors choose the unadjusted p-values. Defined FC for fold change.

Response: Differential gene expression analysis was performed using DESeq with FDR multiple testing correction of p values.

FC was changed to Fold change in line 26. |log2 FC| > 0.58 was changed to Fold change > 1.5 or fold Change < 0.67 in line 95.

Line 45-46: please provide the supporting references.

Response: the references were added.

Line 51-60: as the manuscript involving in a different growth state of the duck, the authors might summarize the growth stage of the duck and how the growth curve of the difference stage looks like.

Response: There is seldom literature involving growth curve of duck till 66 week.

Line 74-75: Why did the authors choose these weeks? Please justifying.

Response: Actually, we chose female duck as our research object. 35 week is nearly the peak of egg production.

Are these animals related?

Response: These ducks were randomly selected and they were not related.

Are the any genetic/genomic selections for these animals or herds?

Response: There were no genetic/genomic selections for these animals.

Line 77-79: As the fat amount is highly related to body weight, please provide the bodyweight of these animals. How did the authors select animals?

Response: we select animals with similar weight. The bodyweight was unrecorded.

Please provide the details of bioinformatic analyses, the software for using the quality control, trimming, alignment, etc.

Response: The details were added as follow. Sequence adapters and low quality reads were removed using Trimmomatic. The quality control of raw sequence was performed by FastX (http://www.bioinformatics.bbsrc.ac.uk/projects/fastqc/). The paired-end reads were aligned to the duck reference genome using the Tophat2 version 2.0.11 (http://ccb.jhu.edu/software/tophat/index.shtml).

How did the authors perform DE analyses, which software? Did the authors normalize the data, which models used in DE analyses?

Response: To calculate the expression quantity of each transcript, alignment results were analyzed by the Cufflinks version 2.1.1. FPKM (Fragments Per Kilobase of exon model per Million mapped reads) method was used to quantify the gene expression level of all samples. DEGs were identified using the DESeq (2012) R package functions estimate SizeFactors and nbinomTest. P value < 0.05 and fold Change >1.5 or fold Change < 0.67 was set as the threshold for significantly differential expression.

Why did the authors select p-value? The authors should use P values controlling for multiple testing such as FDR?

Response: Differential gene expression analysis was performed using DESeq with FDR multiple testing correction of p values.

Line 100: Give citation for Cytoscape

Response: The citation for Cytoscape was added.

The raw sequence data should be deposited in the public domain for the reproducibility of research.

Response: the raw data for this study was submitted to the Genome Sequence Archive in BIG Data Center, Beijing Institute of Genomics (BIG), Chinese Academy of Sciences, under project number PRJCA005390 and accession number CRA005802. Now it is under checking.

Please avoid the abbreviation in the table and figure titles.

Response: done.

Table 1 is not so important. I suggest the authors present the table with the top DE genes for each comparison group.

Response: The top DEGs were added in table 2 and 3.

Line 114: What is the reason for the selection of the log fold change threshold of 0.58 and please define the abbreviation for it

Response: Actually the threshold I used was fold change > 1.5 or fold change < 0.67, which is equal to |log2(fold change)| > 0.585.

I suggested the authors used the Volcano plots instead of the heatmap in figure S1.

Response: Volcano plots could show the distribution of up and down genes. The heatmap could clustered samples with similar expression patterns. Compared with Volcano plots, the heatmap could reveal more information, including the differences and homoplasy of each sample. So I think the heatmap maybe better.

Line 135 and others: Please keep gene names in Italics.

Response: Done

Line 147: To identify the subnetwork, I suggest the authors use another app such as the MCODE app

Response: the subnetwork content by MCODE were added.

Line 172: Change 3 to three and do the same for a number smaller than 11

Response: Done

Line 214-217: it is not necessary to repeat the results.

Response: The repeated part was deleted.

Line 236: please explore the relationship between HSP and fat deposition or lipid metabolism.

please explore the relationship between HSP (heat shock proteins) and fat deposition or lipid metabolism.

Response: though some references showed the relationship between HSPs and fat deposition. The second reviewer gave me a more reasonable explanation. HSPs respond rapidly to changing environmental conditions to protect the cellular proteins. The ducks were kept for a long time where the environmental conditions might have changed. Therefore, it is likely that the identified differentially expressed HSP genes were a response to these conditions rather than being involved in subcutaneous adipogenesis. So the related description were deleted.

Line 239-240: The assumption is vague and no supporting references.

Response: the supporting references (reference 11 to 13) were given.

Line 242: the sentence is not supporting any things?

Response: Modified.

Line 251-252 Explore more in the mechanism.

Response: It is reported that low vitamin D status was association with obesity status[1]. Vitamin D deficiency in old rats increases adiposity and conditions improved by vitamin D supplementation[2].

Line 258-259: in which species?

Response: in Human.

Line 261: How did the authors explain the link between the fat deposition in duck with the plasma lipid in humans?

Response: on a large scale, the fat deposition in duck and the level of plasma lipid in humans were both related to lipid metabolism. But the link is not close. So I had deleted the content.

References

  1. Savastano, S.; Barrea, L.; Savanelli, M. C.; Nappi, F.; Di Somma, C.; Orio, F.; Colao, A. Low vitamin D status and obesity: Role of nutritionist. Reviews in endocrine & metabolic disorders 2017, 18, 215-225.
  2. Chanet, A.; Salles, J.; Guillet, C.; Giraudet, C.; Berry, A.; Patrac, V.; Domingues-Faria, C.; Tagliaferri, C.; Bouton, K.; Bertrand-Michel, J.; et al. Vitamin D supplementation restores the blunted muscle protein synthesis response in deficient old rats through an impact on ectopic fat deposition. J Nutr Biochem 2017, 46, 30-38.

Reviewer 2 Report

The authors describe a RNA-seq experiment in subcutaneous fat (SCF) of Muscovy ducks at different developmental stages. Each three samples of SCF were pooled from each developmental stage resulting in three biological replicates per stage. This is correct from a methodological point of view. The study revealed 138 genes differentially expressed (DEG) between juvenile and adult stages and 129 DEG between adult and aged ducks. Standard analyses (GO enrichment and KEGG pathway analysis) revealed some terms and pathways enriched in the data set. Finally, protein-protein-interaction networks were constructed. Regarding the definition of DEGs, I have a serious concern:

(I) While the setting of FC =1.5 is adequate, there was no correction for multiple testing applied. Using uncorrected p-values of p<0.05 for definition of significant changes will inevitably lead to a high number of false positive results given the total number of ~ 14.000 mapped genes. Therefore, a correction (e.g. Bonferroni) is necessary.

Another major concern is the interpretation of the data regarding the heat shock proteins (HSPs). The ducks were kept for a long time (66 weeks) where the environmental conditions might have changed. It is well known that HSPs respond rapidly to changing environmental conditions to protect the cellular proteins. Therefore, it is likely that the identified DE HSP genes were a response to these conditions rather than being involved in subcutaneous adipogenesis

Minor:

L116/117: „Moreover, 17 DEGs were both significantly differential expression in both comparisons (Figure 1C).“ Reword, as this is grammatically incorrect.

L308 et seq.: The reference list must be formatted according to the journals’ style. Now there are many inconsistencies regarding journal abbreviations vs. long versions, upper and lower case mix-up, Korean signs in author list etc.

Author Response

Dear editor,

I quite appreciate your favorite consideration and the reviewer’s insightful comments. Now I have revised the manuscript according to the reviewer’s comments. The revisions were addressed point by point below.

Reviewer 2:

(I) While the setting of FC =1.5 is adequate, there was no correction for multiple testing applied. Using uncorrected p-values of p<0.05 for definition of significant changes will inevitably lead to a high number of false positive results given the total number of ~ 14.000 mapped genes. Therefore, a correction (e.g. Bonferroni) is necessary.

Response: Differential gene expression analysis was performed using DESeq with FDR multiple testing correction of p values.

Another major concern is the interpretation of the data regarding the heat shock proteins (HSPs). The ducks were kept for a long time (66 weeks) where the environmental conditions might have changed. It is well known that HSPs respond rapidly to changing environmental conditions to protect the cellular proteins. Therefore, it is likely that the identified DE HSP genes were a response to these conditions rather than being involved in subcutaneous adipogenesis.

Response: Thank you. I think your points are very reasonable. So the related description were deleted.

Minor:

L116/117: Moreover, 17 DEGs were both significantly differential expression in both comparisons (Figure 1C).“ Reword, as this is grammatically incorrect.

Response: moreover, 17 genes were both differentially expressed in these two comparisons.

L308 et seq.: The reference list must be formatted according to the journals’ style. Now there are many inconsistencies regarding journal abbreviations vs. long versions, upper and lower case mix-up, Korean signs in author list etc.

Response: Modified.

Reviewer 3 Report

Dear Authors,

I think your paper is interesting and well presented, minor revisions are required. Two are the points to work on: references inclusion and outputs applicability.

Some more references should be reported to the different manuscript sections.

Abbreviations and acronyms should be written completely the first time they are cited.

English language needs some revision.

Simple Summary: in this section genes and phenotypes acronyms could be limiting factors in "simple" summary comprehension, substitute them with concise descriptions of their functions.

Abstract: ok

Introduction: 

46: add reference after "high fat content"

48: add (FCR)

57-60: please rephrase

62: adipose is an adjective, adipose tissue (please correct it throughout the paper)

62: developmental stages 

63-64: please rephrase

Materials and Methods:

add references throughout the chapter

69: samples' collection

73: add an official protocol number/id

78: selected - explain please

79: adipose tissue was collected

Results:

120: adipose tissue

132: again adipose tissue

154: add references

161: abbreviations

167: all above mentioned genes were

Discussion:

199-200: add references 

202: delete "in summary"

205: add the research group author and year of publication 

207: genus and specie in italic please

208: genus in italic please

208: domesticated form of Cairina, cite phylogeny researches

210: genus and specie in italic please

in general: You could try to emphasize and simplify your results considering the high number of informations you report in the five figures you included. 

Conclusions:

Please, considering the significant results you report in this paper, try to describe their application considering selection and management of productive ducks.

Author Response

Dear editor,

I quite appreciate your favorite consideration and the reviewer’s insightful comments. Now I have revised the manuscript according to the reviewer’s comments. The revisions were addressed point by point below.

Reviewer 3:

Simple Summary: in this section genes and phenotypes acronyms could be limiting factors in "simple" summary comprehension, substitute them with concise descriptions of their functions.

Response: Simple summary was rewriting, and genes acronyms were deleted.

Abstract: ok

Introduction:

46: add reference after "high fat content"

Response: The references were added.

48: add (FCR)

Response: Done.

57-60: please rephrase

Response: Modified.

62: adipose is an adjective, adipose tissue (please correct it throughout the paper)

Response: Done.

62: developmental stages 

Response: Modified.

63-64: please rephrase

Response: Done.

Materials and Methods:

add references throughout the chapter

Response: The references were added.

69: samples' collection

Response: Modified.

73: add an official protocol number/id

Response: The permit No. was added.

78: selected - explain please

Response: We chose healthy individuals with similar size.

79: adipose tissue was collected

Response: Modified.

Results:

120: adipose tissue

Response: Modified.

132: again adipose tissue

Response: Modified.

154: add references

Response: These descriptions were my own results, so I didn’t add references.

161: abbreviations

Response: The first “BP”, “CC”, “MF” abbreviations were on line 143, and the full name were given.

167: all above mentioned genes were

Response: Modified.

Discussion:

199-200: add references 

Response: Added.

202: delete "in summary"

Response: Done.

205: add the research group author and year of publication

Response: Added.

207: genus and specie in italic please

Response: Done.

208: genus in italic please

Response: Done.

208: domesticated form of Cairina, cite phylogeny researches

Response: Added.

210: genus and specie in italic please

Response: Done.

Conclusions:

Please, considering the significant results you report in this paper, try to describe their application considering selection and management of productive ducks.

Response: Added.